# A Comprehensive Review on Starch-Based Hydrogels: From Tradition to Innovation, Opportunities, and Drawbacks

**DOI:** 10.3390/polym16141991

**Published:** 2024-07-11

**Authors:** Katerina Koshenaj, Giovanna Ferrari

**Affiliations:** 1Department of Industrial Engineering, University of Salerno, 84084 Fisciano, Italy; k.koshenaj@studenti.unisa.it; 2ProdAl Scarl, c/o University of Salerno, 84084 Fisciano, Italy

**Keywords:** starch-based hydrogels, traditional methods, innovative alternative methods, high-pressure processing, opportunities, limitations

## Abstract

Natural hydrogels based on renewable and inexpensive sources, such as starch, represent an interesting group of biopolymeric materials with a growing range of applications in the biomedical, cosmeceutical, and food sectors. Starch-based hydrogels have traditionally been produced using different processes based on chemical or physical methods. However, the long processing times, high energy consumption, and safety issues related to the synthesis of these materials, mostly causing severe environmental damage, have been identified as the main limitations for their further exploitation. Therefore, the main scientific challenge for research groups is the development of reliable and sustainable processing methods to reduce the environmental footprint, as well as investigating new low-cost sources of starches and individuating appropriate formulations to produce stable hydrogel-based products. In the last decade, the possibility of physically modifying natural polysaccharides, such as starches, using green or sustainable processing methods has mostly been based on nonthermal technologies including high-pressure processing (HPP). It has been demonstrated that the latter exerts an important role in improving the physicochemical and techno-functional properties of starches. However, as for surveys in the literature, research activities have been devoted to understanding the effects of physical pre-treatments via high-pressure processing (HPP) on starch structural modifications, more so than elucidating its role and capacity for the rapid formation of stable and highly structured starch-based hydrogels with promising functionality and stability, utilizing more sustainable and eco-friendly processing conditions. Therefore, the present review addresses the recent advancements in knowledge on the production of sustainable starch-based hydrogels utilizing HPP as an innovative and clean-label preparation method. Additionally, this manuscript has the ambition to give an updated overview of starch-based hydrogels considering the different types of structures available, and the recent applications are proposed as well to critically analyze the main perspectives and technological challenges for the future exploitation of these novel structures.

## 1. Introduction

The constantly growing world population and international sustainability agreements have encouraged industries and academia to individuate and develop sustainable processes and products beyond revenue; thereby facing the economic, social, and health imbalances that humanity is experiencing. Among the international strategies within the United Nations’ 2030 goals, the development of plant-based systems in view of replacing or reducing the utilization of synthetic materials has received significant attention. 

Among them, hydrogels, which represent a group of versatile polymeric structures, when produced from natural sources such as proteins, starch, cellulose, and gums, among others, could contribute to the development of bio-based materials, resembling live tissues, to be used in biomedical, cosmeceutical, pharmaceutical, and food applications [1]. In particular, starch is one of the most promising biomaterials due to its low cost, abundance, excellent biodegradability, and availability compared to other polymers [2]. Moreover, starch properties such as pasting properties, retrogradation, thermal properties, digestibility, rheological characteristics, swelling capacity, solubility, and water absorption [3] are fundamentals in hydrogel formulation.

Traditionally, they are produced by different chemical or physical methods. On the one hand, chemical methods using cross-linking agents, such as citric acid, glutaraldehyde, sodium trimetaphosphate (STMP), and 1-ethyl-3-(3-dimethyl-aminopropyl-1-carbodiimide) (EDC), or polymerization methods, based on acrylamide and acrylic acids, are commonly used to form chemically cross-linked or grafted starch-based hydrogels [4]. On the other hand, physical environmentally friendly methods, such as heat gelatinization and high-temperature extrusion, are commonly utilized to form physically “non-permanent” starch-based hydrogels that are widely used in the food industry to produce staple foods [4]. Nevertheless, these methods have shown important limitations such as environmentally harmful processing conditions and high energy consumption together with the unstable mechanical properties of the structures formed [5]. Therefore, in the last decade, the scientific community have investigated alternative technologies and suitable processing conditions allowing us to overcome the limitations of conventional methods and obtain starch-based hydrogels characterized by high biodegradability, functionality, and safety. 

Among the alternatives, high-pressure processing (HPP), which is mainly used at the industrial scale to extend the shelf life of food products in the absence of preservatives and additives with no or minimal sensory and nutritional properties damaging the products, has been selected. Indeed, it is well known that HPP can cause physicochemical transformations of biopolymers in certain temperature ranges, through the formation of non-covalent bonds, ionic bonds, and hydrophobic bonds, denaturation, or gelatinization [6,7,8,9]. 

HPP has been proven effective for the physical modification of starch in aqueous suspensions and for the production, at certain processing conditions, of highly structured and functional hydrogels with suitable stability and functionality [10,11,12,13,14]. Several factors, including the starch source, starch/water ratio, processing time, pressure, and temperature, can significantly affect the production of starch-based hydrogels under pressure [15].

In the last few decades, starch-based hydrogels have been highlighted as an important group of sustainable materials to be potentially used for several applications such as dye sorption and metal capture, as well as in agriculture, electrical systems design, and food preparation [4,16]. Moreover, the use of HPP for inducing starch modification, gelatinization, and subsequent starch retrogradation has been recently discussed [17,18]. 

This paper aims at addressing all the relevant, interlinked factors of starch-based hydrogels’ production via HPP, including starch gelatinization occurrence and the physical characterization of the structures formed, and discusses the recent approaches to increase the sustainability of the overall process. The most relevant and recent applications of starch-based hydrogels to predict the potential uses of HPP starch-based hydrogels have been also reviewed, and the main scientific challenges for the exploitation of HPP as a green method for the preparation of these novel structures and their future perspectives have been discussed.

## 2. Starch-Based Hydrogels

### 2.1. General Overview

The demand for plant-based materials, based on the healthy lifestyles stated by governmental policies, has increased over time and become an important mission of many companies around the world. Likewise, bio-based products have emerged to replace or reduce the utilization of synthetic materials to fulfill consumer demand for natural products in all industrial sectors. Among bio-based products, hydrogels play an important role in designing and producing cosmetics, drugs, foods, and biomedical products. These structures are prepared from natural biopolymers with specific structural and mechanical properties, or they are synthesized from agri-food by-products [19,20]. 

Hydrogels based on renewable natural polymers, such as cellulose, polysaccharides, and proteins, have been produced with various traditional methods and popularized for different applications [1,21]. Due to the health and environmental issues related to the production of synthetic polymer-based hydrogels, natural compounds-based hydrogels have received significant attention in recent years as suitable alternative bio-based structures to replace the synthetic ones [22,23,24,25,26,27,28,29,30,31,32,33,34,35,36,37,38,39]. Among green hydrogels, the starch-based ones are the most promising alternatives for producing biopolymers due to their excellent biocompatibility, biodegradability, and potential applicability in numerous sectors [4]. Indeed, according to PubMed data science, research activities on starch-based hydrogels have shown growth with the number of published articles in 2024 being approximately 21 times the number in 2014, highlighting the scientific interest in these novel structures [4,7,8,40,41,42,43,44,45,46,47,48,49,50].

However, several challenges must be considered to ensure their reliable production and safe utilization.

### 2.2. Historical Background

The first colloidal gel preparation with organic salts dates back to the end of the XIX century [51], and the very first successful application of hydrogels was for the manufacturing of contact lenses [52]. The latter boosted the utilization of hydrogels as a group of smart materials with vast applicability. 

According to Buwalda et al. [53], the research activities on hydrogels in recent years were devoted initially to the preparation of these structures utilizing different polymeric sources. Therefore, research efforts were made for the preparation of hydrogels capable of responding to specific stimuli such as pH and temperature, as well as on the effects of biocompounds’ concentration on gel formation or the study of drug release from these structures. Finally, the research activities were focused on the production of tailor-designed hydrogels with tunable characteristics by cross-linking methods with special affinity to the human body [53]. Recently, the need to investigate new methods to produce natural and sustainable hydrogels with defined properties and functionality has arisen. Although the incorporation of natural polymers, such as starch, into hydrogel formulations was proposed at the end of the 1930s in the last century [54], the use of different starches for the production of hydrogels was proposed only in the last few decades, due to the increasing consumer demand for greener and sustainable materials [1]. Many authors investigated different synthetization routes to individuate the most suitable methods for starch-based hydrogel production. Heller et al. [55] utilized a grafted polymerized starch solution to produce hydrogels in the presence of an unsaturated acid, and the structures obtained were proposed as self-regulating drug delivery systems. In the same line, Pereira et al. [56] developed biodegradable hydrogels, based on corn starch/cellulose acetate blends, produced by free-radical polymerization as alternative carriers for drug delivery. Following these investigations, starch-based hydrogels were extensively produced and proposed for a wide range of applications [15,35,57,58,59,60,61,62]. Moreover, in the last few years, the increasing attention of consumers towards materials’ origin and environmental issues has encouraged the scientific community to carry out research activities on the utilization of innovative and “clean-label” technologies to produce starch-based hydrogels [25,63,64]. Figure 1 summarizes the research progress on starch-based hydrogels over recent years based on findings from the literature. 

### 2.3. Classification

The ways the starch-based hydrogels can be classified are vast due to the different aspects involved in their production and tailor-design properties. Considering the huge number of structures defined as hydrogels based on starch as a natural polymeric source and the lack of consensus regarding their classification, an updated classification of starch-based hydrogels is proposed in this work and reported in Figure 2. This classification, based on findings from the literature, considers source, physical appearance, physical stability, composition, methods of preparation, digestibility, and the field of application.

### 2.4. Source

According to the origin of starch as a raw material for their preparation, starch-based hydrogels can be classified as conventional or non-conventional [65]. The major conventional starches include corn, potato, cassava, and wheat, which dominate the current market and have diverse applications in different areas [66]. Due to the recent growing demand for starches, increasing research has been focused on non-conventional starch from a variety of sources, thus expanding the range of its potential application [67,68]. Additionally, the feasibility of using non-conventional starches as renewable materials for commercial applications may reduce the cost of industrial raw materials. Non-conventional starch sources mainly include unripe fruits, rhizomes (ginger, turmeric, and lotus), cereals (amaranthus and millet), and nuts, along with various agricultural waste products and by-products of fruit or vegetable processing. A current overview of the recently investigated starch recovery from unconventional sources, as well as their features, applications, future trends, and limitations, is reported by [69]. Furthermore, Akhmad et al. [70] produced hydrogels using pectin and starch recovered from banana peels, and Jucilene Sena dos Santos et al. [71] produced hydrogels using non-conventional starches from guabiju, pinhao, and uvaia seeds, which showed promise in agriculture applications.

### 2.5. Physical Appearance

Based on the polymerization method applied for their production, hydrogels might appear as a matrix, film, or microsphere [72]. Moreover, hydrogels produced by high-pressure processing (HPP) could display cream-like and rubber-like mechanical properties when cereal (wheat, rice, and corn) or root (tapioca) starches are used as polymeric sources [7]. 

### 2.6. Physical Stability

Hydrogels can be divided into two categories based on their physical stability. Hydrogels obtained from physical methods are considered reversible due to their non-permanent bonds. This hydrophilic polymeric network is typically formed by either a physical entanglement of the polymer chains or non-covalent interactions, including self-assembly through hydrogen bonds, van der Waals interactions, hydrophobic interactions, ionic interactions, and ionic forces, among others [1]. Due to their characteristics, physical hydrogels are considered weak gels with poor mechanical properties compared to chemical hydrogels, this being the main disadvantage of these structure and a challenge for further future development for the scientific community. Nevertheless, a growing interest in physical or reversible gels has been observed due to advantages in their preparation, such as ease production, absence of cross-linking agents, and reversibility [72].

Chemical or irreversible hydrogels are mainly obtained through covalent cross-linking that has a permanent junction [1]. Chemical cross-linking includes grafting monomers to the polymer’s backbone or linking the chains of two polymers together with a cross-linking agent [73]. For chemical hydrogel preparation a monomer, a cross-linker, and an initiator are required, and, in some cases, unwanted toxic by-products are formed [74]. 

### 2.7. Composition

Typically, pure starch can be used for the preparation of starch-based hydrogels. The weak mechanical properties of these structures have increased the demand of smart strategies to increase their mechanical properties, such as the incorporation of other compounds in hydrogel formulations. In the last decade, the preparation of hybrid or composite hydrogels by different cross-linking approaches or by the addition of nanomaterials, inorganic fillers, or nanofibers in the matrix, as well as the combination of polymeric sources has been proposed as an alternative preparation method, in order to enhance the mechanical properties and the water content of hydrogels, as well as to expand the range of applications [1,75]. For instance, PVA-based hydrogels are one of the first hybrid hydrogels tailored for tissue engineering. Polyvinyl alcohol (PVA) is a hydrophilic polymer with reduced biocompatibility, biodegradability, and polar solubility, and in a polymerized form it displays rigidity, poor adhesiveness, and certain levels of cytotoxicity [76]. Additionally, when blending two polymers, the processability must be carefully considered. Films made from starch/PVA blends are typically produced by gelatinizing starch in the presence of a PVA polymer, which cannot be melt processed because its thermal degradation temperature is slightly higher than its melting temperature. Moreover, interfacial adhesion is another crucial aspect that must be considered. The mechanical properties of simple mixtures of starch and polycaprolactone (PCL) are considerably poorer than pure PCL. Such composites do not exhibit any significant interfacial adhesion and in order to enhance this property, structural modifications have been applied to target the desired applications [76]. It has been found that the utilization of fillers in a starch matrix is an effective method to obtain high-performance starch-based composites. Moreover, the utilization of natural fillers provides positive environmental benefits in terms of ultimate disposition and raw material use [77]. The key elements that determine the physicochemical properties of the final composite are the filler size and interfacial adhesion between the filler and the matrix. Mechanical bonding, which is caused by the roughness of both surfaces, is a type of bonding that can improve the filler–matrix interaction at the interface. Based on recent reviews, hybrid hydrogels may exhibit antibacterial and osteogenic activities and are particularly beneficial in biomedical engineering [40,41,78]. Moreover, Piola et al. [79] developed a cross-linked 3D-printable hydrogel based on biocompatible natural polymers that could be used as a matrix for cell growth and for the development of in vitro tissues, as well as for wound dressing. 

Hydrogels typically lose their thermal stability when exposed to high temperatures. As a result, enhancing the thermal stability of hydrogels has become an area of significant interest. One emerging strategy to achieve this is the development of hydrogel nanocomposites. These nanocomposites are formed by combining biopolymers, such as polysaccharides, polypeptides, proteins, aliphatic polyesters, and polynucleic acids, with various fillers including clays, hydroxyapatite, and metal nanoparticles [80]. Incorporating nanoparticles into both natural and synthetic polymer hydrogels imparts unique physicochemical properties, such as improved mechanical strength, thermal stability, sound absorption, optical clarity, magnetic responsiveness, electrical stimulation, selectivity, and high swelling rates. This enhancement significantly broadens the potential applicability of these advanced materials [81,82].

### 2.8. Digestibility

The digestion of different components of starch-based hydrogels takes place at a different rate in the gastrointestinal human tract. The salivary α-amylase enzyme initiates digestion in the oral cavity, where it continues under the influence of pancreatic α-amylase and α-glucosidase enzymes in the intestine, after being exposed to stomach conditions [83]. The digestibility of starch-based hydrogels strictly depends on the starch source used in the formulation. From a nutritional perspective, starch is classified as rapidly digestible (RDS), slowly digestible (SDS), or resistant starch (RS) [84]. RDS is rapidly digested and absorbed in the duodenum and proximal regions of the small intestine, causing a rapid elevation of blood glucose. SDS is digested slowly in the small intestine, while RS is not digested and undergoes fermentation in the large intestine [85]. Several factors can affect the digestibility of the starch, highlighting the amylose/amylopectin ratio as the most important. Generally, starches with high amylose content tend to be more resistant to hydrolysis in the small intestine [86]. An important step of hydrolysis is the diffusion of α-amylase into the substrate. Previously, it was assumed that the hydrolysis of starch starts from the surface of the granules. However, it was found that native cereal starches possess peripheral pores and channels, allowing α-amylase to enter the granules, resulting in an inside–outside hydrolysis mechanism [87]. In contrast, potato and other B-type starches are digested from the surface of the starch [88] and this can explain the higher digestibility of cereal starches compared to tuber starches [87]. Consequently, starch-based hydrogels are considered an appropriate target for oral, gastric, and intestinal applications [42,89,90].

### 2.9. Applications

Starch-based hydrogels have been produced, patented, and commercialized for use in products such as hygiene products, tissue engineering scaffolds, drug delivery systems, wound dressings, and agricultural materials, as summarized in Table 1.

In the last decade, the applications of these structures have become more prevalent in food and agricultural sectors, as shown in Figure 3. Agriculture is one of the most crucial industries that relies on an adequate quantity of water and nutrients that are provided by fertilizers, which in turn cause excessive environmental pollution. The utilization of hydrogels in farmlands has several advantages, including the need for less water for irrigation, reduction in plant mortality, increase in plant growth, and reduction in environmental pollution. In the food industry, starch-based hydrogels are primarily used for food packaging, thereby reducing the use of petroleum-based plastics. On the contrary, their use in other industries, such as biomedicine and cosmetics, has faced several limitations, such as causing negative environmental impacts and having weak mechanical properties [20,43,91].

(a)Biomedical applications

The use of starch-based hydrogels in biomedical applications as biodegradable and biocompatible materials is highly recommended. Recently, Qamruzzaman et al. [16] reviewed a number of potential possibilities for starch-based hydrogels in the biomedical field, particularly in tissue engineering and drug delivery systems. Modified starch hydrogels have shown promising ability to encapsulate and release drugs at physiological pH and temperature, which are determined by their specific composition [44,45]. Moreover, the increased use of starch-based hydrogels in drug release systems has enhanced the interest on these materials at the industrial level. Starch-based hydrogels are promising candidates for tissue engineering due to their well-known ability to retain a substantial amount of water. However, only few authors reported their use in this field. Starch-based physical hydrogels have been produced by adding to the formulation chitosan and polyvinyl alcohol, and the results in terms of structure, swelling, and safety in vitro and in vivo demonstrated their potential use in tissue engineering [46]. In addition, Cui et al. [47] developed in situ enzyme-linked starch-based hydrogels with promising use as tissue adhesives and homeostatic agents. The synthesis of hydrogels used for contact lenses belongs to the first generation of these materials and has gained success in the market as a material for soft contact lenses. Although edible polysaccharide starch has been suggested as a smart material, investigations of their use in contact lenses are limited. Further studies are needed to expand the possibilities of applying starch-based hydrogels in the biomedical sector. Therefore, the challenge for scientists is to modify the structures of starch-based hydrogels and improve their mechanical properties and stability. The key to obtaining hydrogel structures with desirable mechanical properties is the relation between the structure and the construction methods. Zhang et al. [92] reported strategies to fabricate starch-based hydrogels, which include the utilization of conventional methods, such as graft polymerization, chemical modification, and radiation-induced polymerization, highlighting the advantages and limitations of each of them. Furthermore, it has been shown that a double network method, where starch is the primary network compound and polyvinyl alcohol the secondary one, could enhance the mechanical properties of starch hydrogels [93]. Another possible way to improve mechanical properties is the incorporation of micro- and nanosized fillers into starch materials. Nanofillers are much more promising than microfillers due to the mutual effect between the fillers at the nanometer scale, allowing them to form molecular bridges in the starch matrix and resulting in an improvement in nanocomposite properties, such as mechanical properties, thermal stability, moisture resistance, oxygen barrier properties, and biodegradation rate [94,95,96].

(b)Cosmetic applications

Hydrogels combined with natural or synthetic compounds may be used in cosmetic products for external use on the body and hairs, with a prolonged stay on the application site and a reduced need for the product’s increased delivery frequency. Over the past decade, natural polymers reduced the use of synthetic materials in cosmetic formulations. Mitura et al. [91] highlighted the potential benefits of studying hydrogels’ production based on biopolymers for the cosmetic industry. Owing to their physical properties and easy functionalization, starch-based hydrogels can be used as smart carriers to encapsulate bioactive compounds that can be applied in the production of cosmeceuticals, nutraceuticals, and nutricosmetic products. Encapsulation has been suggested to improve stability and resistance against degradation, as well as to control the release of active ingredients used in cosmetic products [97]. Recently, Pagano et al. [48] produced starch-based hydrogels using saffron stigma, which is widely used in cosmetics as an anti-age and anti-UV agent, and as a smelling agent for perfumes. The obtained hydrogels exhibited good rheological properties and spreadability, which are necessary for skin applications. An effective strategy for achieving efficient compound release in cosmetics is the development of adhesive hydrogels that allow longer residence times at the application site, maintaining a high local concentration of the active ingredient [49,98,99,100]. Hydrogels can achieve strong adhesion with a surface that is applied through bonding mechanisms such as mechanical interlocking, diffusion, and wet adhesion [50,98]. Recently, D’Aniello et al. [50] encapsulated green tea extract into starch-based HPP hydrogels and obtained a controlled release over time of the bioactive compounds, highlighting their effective incorporation inside these structured materials. The authors suggested their utilization as a good starting point for the design of topical products. However, the utilization of starch-based hydrogels in cosmetics has not been thoroughly exploited, and further research efforts and knowledge improvements are required.

(c)Agriculture applications

It is well known that hydrogels can absorb and retain large amounts of water or aqueous solutions in their structure without dissolving. They can gradually release up to 95% of water in a dry environment while they rehydrate when they are exposed again to a wet environment [101]. This behavior makes hydrogels an interesting structure to be used in agriculture, primarily as water-retention agents for soil conditioning, or as carriers of agrochemicals for their gradual or continuous release [102,103]. One of the key factors restricting the growth and productivity of crops and fruits is the ability of plants to survive in irrigation systems used to prevent water stress. In this regard, natural polymers could be candidates for enhancing the water retention ability of soil [104]. Furthermore, bio-based superabsorbent hydrogels, which can absorb and retain aqueous solutions up to hundreds of times their weight while maintaining their network, could easily find application in agriculture [105,106]. Several physical treatments and chemical agents, like initiators, are utilized to prepare superabsorbent hydrogels [107,108,109,110]. Table 2 lists the literature’s findings on the synthesis of starch-based hydrogels for agricultural applications. The structural stability and swelling behavior of starch hydrogels are key aspects that indicate their suitability for these applications, and they are primarily controlled by the properties of the hydrogel network and the synthesis methods [111]. Likewise, the bulk of published studies are generally focused on enhancing the swelling properties of starch hydrogels, primarily by optimizing their formulations. Although starch-based hydrogels are widely used in agriculture as smart strategies that offer many benefits to industry and the environment, there are still some drawbacks to their synthesis. The main limitations are linked with the large-scale production of starch-based hydrogels by conventional technologies; hence the development of sustainable strategies is necessary for enabling further advancement in this field. The main challenges include developing novel environmentally acceptable processes, particularly for chemically cross-linked hydrogels, and producing low-cost hydrogels with great absorption capacity and appropriate mechanical properties.

(d)Food applications

In the food industry, starch-based biopolymers are primarily used in food packaging. The requirements for food packaging include minimizing food loss, maintaining food freshness, increasing organoleptic characteristics, and ensuring food safety [121]. Although starch is one of the best alternatives to synthetic polymers, its application is limited because of undesirable properties, such as retrogradation and brittleness [122,123,124]. In this regard, various humectants such as glycerol, urea, formamide, amino acids, sorbitol, and citric acid have been added to increase the flexibility of starches in packaging [125,126]. Moreover, starch and other polymers may be blended to achieve high chain flexibility for water transport, which is required for the application of hydrogels particularly in the food preservation field. The blending capacity occurs from molecular modifications through the reactions of functional groups (amino and hydroxyl), which provide hydrogels with various possible structural and morphological profiles, increasing their potential of absorption and liquid retention capacity [127,128,129].

It should be highlighted that biodegradable food packaging is mostly focused on applications for products with a short shelf life and dry foods [130]. However, starch-based hydrogel utilization in the food industry is not limited to packaging. They are applied for the production of starch noodles and are carriers of active ingredients [4]. An example of starch-based hydrogels can be found in the production of wet noodles, where gelatinization and cooling-induced retrogradation of starch occur [131]. When used as carriers of food-active ingredients, they have demonstrated an effective encapsulation capacity [132]. However, these applications were studied only at the laboratory scale. Additional studies are required to investigate the effectiveness of starch hydrogel encapsulation for the controlled release of compounds, considering that the main factors significantly affecting their functional properties are starch structures and starch modifications.

### 2.10. Methods of Preparation

The synthesis of starch-based hydrogels can be accomplished by different traditional and innovative methods. 

### 2.11. Traditional

Traditional methods, which include physical and chemical ones, are further divided into synthetic and green categories due to their effects on the environment and human health. Thermal treatments (heating/cooling), freeze–thawing processes, ultrasonic treatments, hydrogen bonding, and complex coacervation are the most used physical methods to produce hydrogels, as reported in the literature. Among the traditional methods thermal treatments allow us to induce the complete gelatinization of a polymeric source and obtain homogeneous materials with smooth surfaces and good processability [133]. Thermal hydrogel formation occurs due to helix development, the association of helices, and the setup of junction zones, as described by Funami et al. [134]. In the freeze–thawing process, instead, the gelation of polymers is caused by physical cross-linking. Freeze and thawing cycles applied to a polymer aqueous solution induce the formation of polymer microcrystals [135]. This method is driven by phase separation and cycling, which occurs as the solution freezes and the polymer is rejected from the growing ice crystallites, and the crystalline structure generated grows as the number of freeze–thawing cycles increases [136]. Natural polymer modification assisted by ultrasonic waves is a simple, eco-friendly, and cost-effective method extensively used in the last few years [137,138]. Ionotropic hydrogels are complex physical gels formed by mixing anionic and cationic polymers. Calcium alginate coacervated with alginate-poly(lysine) to stabilize the structure is a typical example of physical hydrogels produced by complex coacervation [139]. Physical hydrogels can be obtained by H-bonding, consisting of the formation of hydrogen bonds between the polymeric source and water, favoring the formation of the gel network. H-bonded hydrogels are synthesized by changing the conditions of the suspending medium in the presence of a polymeric source of hydrophilic nature. Hydrogen bonds form when the positive hydrogen atom establishes an electrostatic link with electronegative acceptor atoms such as oxygen, nitrogen, or fluoride [140]. An example of H-bonding hydrogel preparation is the production of physical carboxymethyl cellulose (CMC) hydrogels, as was reported by [141].

Chemical methods are extensively used for increasing the physical stability of hydrogels, creating permanent structures. Among them, click chemistry is one of the most promising for producing cross-linked hydrogels thanks to its high specificity, high yield, biorthogonality, and mild reaction conditions [142,143]. Chemical reactions such as Diels Alder, Azide-Alkyn Huisgen, Thiol-ene Photocoupling, and Aldehyde-Hydrazide Coupling have been categorized as click chemistry methodologies that make the production of functional hydrogels easier [142]. Injectable hyaluronic poly-ethyleneglycol (HA-PEG) hydrogels, 3D-patterned hydrogels and injectable HA-pectin-based hydrogels have been produced by click chemistry methods for biomedical applications [143,144,145,146]. Furthermore, grafting polymerization consists of generating free radicals onto a more robust monomer that polymerizes, creating a chain of monomers covalently bound to the support [73]. The grafting process, where the polymer chains are activated, can be produced by chemical agents or radiation treatments. The major advantage of the radiation initiation over the chemical initiation is the production of relatively pure and initiator-free hydrogels [124]. Kowalski et al. [147] reported that superabsorbent hydrogels can be produced by grafting acrylic acid onto carboxymethylated high-amylose corn starch. Moreover, Musa et al. [148] suggested that starch-graft-acrylamide hydrogels can be a potential vehicle for oral drug delivery. Starch-based hydrogels produced by chemical methods show better mechanical properties than those obtained by physical methods. Ozonization chemically modifies starches. The molecular weights of amylose and amylopectin are reduced after ozonization, and the oxidized starch can form starch-based hydrogels with high strength and good rheological properties [149,150]. Lastly, chemical cross-linking is another technique that involves the interaction of new molecules with cross-linker agents in polymeric chains to form a cross-linked network.

Typically, for this kind of preparation, the main target groups of the biopolymers are OH, COOH, and NH_2_ with cross-linkers presented in Table 3. Although chemical hydrogel durability and stability are two desirable characteristics, these positive characteristics have been considered disadvantages in the last two decades. In fact, rigidity, toxicity, and the hydrophobic aggregation of the cross-linking agents (high-density clusters), as well as the gel network defects due to the presence of free chains, have been identified as important drawbacks that reduce the applicability of these structures [1]. 

### 2.12. Innovative

The limitations arising from utilizing traditional methods for hydrogels formation have increased the research interests to find novel suitable alternatives. In the last few decades, several innovative processes to produce hydrogels have been investigated including enzymatic cross-linking [157], ultrasonication [158], and photo-cross-linking [159]. Moreover, one of the most promising strategies for the production of starch-based hydrogels under sustainable and clean-label processing conditions relies on the utilization of high-pressure processing (HPP) as an effective starch modification and gel formation technology. It is well known that HPP promotes sol/gel transitions in proteins and other food components [6]. For this reason, this physical method has been proposed to modify or gelatinize suspensions of different types of starch, allowing us to overcome the limitations of traditional gelation processes such as the long operation times, high energy consumption, and the use of hazardous materials [160]. Therefore, an emerging group of new biomaterials, the starch-based HPP hydrogels, has been recently studied. 

## 3. Starch-Based HPP Hydrogels

### 3.1. High-Pressure Processing (HPP)

High-pressure processing (HPP) has been proposed to produce starch-based hydrogels to overcome the problems arising from the utilization of traditional methods [161]. HPP has been proposed as a nonthermal technology for food sanitization that ensures both food safety and preserves the sensorial and nutritional properties of food products. Temperature, pressure, and treatment time are the three processing variables influencing the effectiveness of HPP technology applications, and the individuation of their optimal values allows design robust and efficient processes. Generally, the pressure is transmitted instantaneously and uniformly to a food, hermetically sealed in flexible packages, using a pressurized liquid medium. The pressure is exerted uniformly to the food product, independently from its size and geometry, differently from heating processes, where heat is transferred gradually through the food system. The principal advantages of the use of HPP on starch suspensions to produce hydrogels are as follows: (I) the use of cross-linkers and additives is avoided, (II) there are no changes in their physical and chemical properties, (III) there is a reduction of processing time, (IV) lower energy is consumed and wasted, and (V) thermosensitive and thermolabile compounds can be added to the starch suspension with limited risks of their damages inside the networks during treatments. HPP technology can be seen as an environmentally friendly processing method [162].

### 3.2. Impact of High Pressure on Starch

High-pressure processing (HPP) causes the disordering of biopolymers, including proteins and starches, which induces modifications of non-covalent intermolecular interactions, including pressure-assisted gelatinization. Under high pressure, starches undergo morphological and structural changes and different gelatinization extents are observed depending on starch sources and composition [163]. Intact or limited swollen starch granules could be observed after treatment, and the HPP starch-based hydrogels formed show different rheological behavior compared to the thermally treated ones. It is worth noting that HPP-assisted gelatinization refers to the modification of intermolecular bonds between starch molecules, allowing the hydrogen bonding sites to engage water into the granules [164], whereas HPP-assisted gelation refers to the formation of a stable gel structure due to polymer branching, facilitated by the compressing forces. A starch gelatinization mechanism under high pressure was proposed by Knorr et al. [6] and Yamamoto et al. [163], as illustrated in Figure 4. During gelatinization under pressure, the amorphous and crystalline regions of the granules are hydrated, and the swelling of the starch granules occur. The smectic crystalline structure is decomposed by helix–helix dissociation, followed by helix coil transition and helix reorganization. Thus, under high pressure, the disintegration of the starch macromolecule is incomplete due to the role of the helix conformations [14]. Moreover, B-type starches, such as potato starch (Table 4), are more resistant to pressure than A- or C-type starches. The higher number of water molecules (36 molecules in the B-type cell compared to only 8 in the A-type elementary cell) could cause the higher-pressure resistance of B-starches. 

Katopo et al. [10] suggested that for B-type starches, water fills up the channel in the cell unit of the crystallite and stabilizes the structure. In contrast, the A-type crystallite has a more scattered amylopectin branching structure, which is more flexible, and therefore allows the rearrangement of double helices to generate a channel in which water molecules are included under pressure. Consequently, the crystalline structure undergoes a transformation under pressure from A-type to B-type crystallites. Tester et al. [86] and Sarko et al. [165] have illustrated the A-type and B-type conformation, as illustrated in Figure 5. Wang et al. [166] reported that lotus seed starch is C-type, and under 600 MPa for 30 min it changes to B-type starch. Similar changes in the diffraction pattern from C-type to B-type were reported for mung bean, lentil, and pea starches by [12,72,167]. According to the type of starch used attached to their structural differences, HPP hydrogels may show creamy or gummy structures [7]. Tapioca (C-type) showed a gummy appearance, which can be related to the lower number of granules per gram of starch compared to corn, rice, and wheat (A-type) [168].

Furthermore, B-type starches, such as potato starch, are characterized by high resistance to pressure-induced gelation, so that very high-pressure levels are required to obtain complete gelatinization. However, Larrea-Wachtendorff et al. [8] showed that high-pressure treatment is an effective method for producing potato starch hydrogels under economically feasible conditions, with processing times well below those used in conventional preparation methods, provided that rice starch fractions with small particle sizes are selected. 

The complex mechanism of starch gelatinization under pressure is affected by numerous factors, such as starch source, starch/water ratio, pressure level, and processing time, as well as processing temperature. The pressure range in which the sol/gel transition under pressure occurs depends on the starch source and crystalline structure (e.g., AM/AP ratio) [11]. Also, the starch concentration plays an important role in the gelation process by pressure, as a good interaction between starch and water is a key factor for obtaining stable gels with almost completely swollen granules [10]. Also, the temperature plays a crucial role with a direct relationship with the degree of gelatinization and the time required to achieve complete gelation by pressure. In fact, at 200 MPa, starch gelatinization occurs at lower temperatures, while the starch/water solutions need to be heated at a higher temperature if the pressure process is carried out at 0.1 MPa [169]. In addition, it is well known that in starch/water solutions, to initiate the gelation process during the pressurization period, an energy equilibrium must be attained but a nonlinear behavior of this phenomenon with time is observed. In this regard, Buckow et al. [170] demonstrated that at 650 MPa and treatment times between 1 and 20 min, the gelation and the maximum swelling was completed after 5 min of treatment and no differences were observed with the samples treated for 10, 15, and 20 min. According to Larrea-Wachtendorff et al. [7], HPP treatments at 600 MPa for 5 and 15 min can be used to obtain stable starch-based hydrogels. The mechanical properties of the hydrogels produced at 600 MPa were improved by utilizing processing times of 15 min. However, this processing time had detrimental effects on the color of the hydrogels, particularly lightness (L*) and whiteness (WI).

The pressure gelatinization of starch can be detected by observing the samples under polarized light. Starch granules show hilum-centered birefringence, which refracts light in an anisotropic material in two slightly different directions to form two rays and corresponds to crystallinity, while gelatinized granules lose the hilum and the birefringence [163]. According to many studies, there were no notable changes in the birefringence pattern on samples treated at lower pressures (150–300 MPa), although there were some losses of the birefringence pattern and crosses at medium pressures (450 MPa) and a full loss of birefringence at 600 MPa, indicating a complete gelatinization [171,172,173]. In Table 5, the studies carried out in the last 30 years on the utilization of HPP treatments to gelatinize or modify starch from different sources as well as the main findings are reported.

A number of researchers have thoroughly reported the impact of pressure on the physicochemical characteristics of starch and starch suspensions [165,167,175,176,183,184]. One of the most important physicochemical properties of starch suspensions under HPP treatments is the swelling capacity and the solubility of amylose, both reflecting the extent of the interactions between starch chains within the amorphous and the crystalline domains [164]. Despite the importance of the swelling process in many technological applications of starches, there is a limited understanding of the factors that control its rate and extent. Different factors are reported to influence the swelling of starch granules, such as botanical source, amylopectin/amylose ratio, granule size, protein content, lipid content, and ash content [185]. Starches, according to the swelling behavior of their granules under pressure, can be classified into those showing very limited swelling and those characterized by extensive swelling similar to that occurring during heat gelatinization. Starches showing limited swelling can be further classified into those forming pastes with a smooth texture and those forming rigid gels. All legume starches are characterized by a higher amylose content, which gives rise to rigid gels [186]. Pulgarín et al. [187] studied the effect of amylose content on gel formation and concluded that a high amylose content negatively impacts the swelling of corn starch granules during high-pressure processing (HPP). This effect is primarily due to amylose stabilizing the starch’s crystalline structure, preventing the helix from unwinding, and thereby restricting water from entering the granules. Moreover, according to the results of these authors, a lower capacity of starch granules to retain water during storage time was observed, suggesting that the high amylose molecules accelerate the retrogradation of starch gels produced under HPP and hinder the water absorption capacity of the starch granules by forming amylose–lipid complexes.

Additionally, high-pressure treatments alter the enzyme susceptibility of starch to α-amylase and amyloglucosidase, with HPP gels showing more resistant starch (RS) than those obtained with thermal treatments [183,188,189]. The increase in RS content reveals that strong amylose–amylose and amylose/amylopectin interactions occurred during pressure treatments [190]. However, Chen et al. [164] recently published contradictory results regarding the influence of high pressure on starch digestibility, highlighting that the botanical source and effects of retrogradation are the key factors determining these different findings. HPP hydrogels from various starch sources exhibit different structural characteristics influencing their digestibility. Furthermore, the degree of retrogradation after HHP, affected by drying methods such as heat drying, freeze drying, and ethanol air drying, significantly alters RS content. However, it is important to note that inconsistent results may also arise from the limited accuracy of currently available kits for testing the RS content. Even though many efforts have been devoted to unravelling the role of chemical–physical characteristics of starches and HPP processing variables on the gelatinization process, only few investigations were carried out to individuate the potential use of such biopolymers, as illustrated in the papers listed in Table 6.

## 4. Challenges, Future Perspectives, and Conclusions

Starch-based hydrogels are among the most promising green hydrogels to be used as alternative biocompatible materials. An increasing number of studies have been focused on these materials in the last few years due to their peculiar characteristics such as safety, biodegradability, and biocompatibility. Although research interests toward starch-based hydrogels are expanding rapidly, there are still some challenges that require further study efforts.

Conventional starches have demonstrated their ability to produce starch-based hydrogels for a wide range of applications, as comprehensively discussed in this review. However, the growing demand of starch in different industrial sectors drives the search for new non-conventional sources of starches alternative to the conventional ones. However, proper extraction and purification processes should be set up and applied to recover non-conventional starches such as those deriving from the valorization of discarded biomasses, such as non-edible parts of fruits, rhizomes, cereals, legumes, and nuts. 

In recent decades, alternative methods for hydrogel formation have been investigated to overcome the limits of the traditional ones, including high-pressure processing. The papers found in the literature demonstrated that high pressure is a valid alternative technology to produce starch-based hydrogels, and the factors controlling the process have been individuated. Moreover, some studies reported the results of HPP gelatinization of several starches or proposed the utilization of HPP as a pre-treatment stage to achieve the modification of the starch microstructure. This notwithstanding, to fully unravel the gelation mechanism under high-pressure processing, further investigations are still needed. More work is also necessary to shade light on the fundamental aspects of sol/gel transition under pressure, and to better clarify the effects of processing variables and starch physical–chemical properties on the production of starch-based hydrogels under pressure. 

Moreover, since it has been demonstrated that starch-based HPP hydrogels are characterized by reduced rheological properties compared with those obtained by chemical methods, further research efforts would be highly desirable to individuate the processing conditions, starch modification strategies, and the addition of technological co-adjuvants to enhance the structural and mechanical properties of starch-based hydrogels in view of their future exploitation. Native starch can be modified to improve its physicochemical properties and structural characteristics, and to increase its functionality. The physical modification involves the treatment of native granules, including temperature, pressure, shear, and irradiation, to enhance water solubility and reduce the size of the starch granules. The chemical modification of starch involves the addition of functional groups to the starch molecule, without affecting the morphology or granule size distribution, to increase the degree of polymerization in the starch granules, modify its solubility in organic solvents, and reduce its swelling capacity. Another strategy for overcoming these mechanical drawbacks is the addition of humectants, which should be small molecules, polar, hydrophilic, and appropriate for the starch polymer. The effectiveness of the most used humectants, such as glycerol, urea, formamide, amino acids, sorbitol, and citric acid, depends on the starch source. In addition, blending starch with other polymers or fillers is an effective practical strategy that has attracted interest, particularly when combined with biodegradable materials to retain the eco-friendly nature of hydrogels. The blending mechanism and its impact on the mechanical properties of starch-based hydrogels need more research.

To conclude, starch-based HPP hydrogels represent an emerging field of research with the main goal of scaling up their preparation process for widespread industrial use. The key challenge will be to maintain a sustainable and eco-friendly product with appropriate structural and mechanical properties. Moreover, further research should be performed on standardizing the physical and microbiological stability of starch-based HPP hydrogels that are principal in the view of optimizing processing conditions, improving the design of these biomaterials, and proposing their utilization in different industrial sectors.

## Figures and Tables

**Figure 1 polymers-16-01991-f001:**
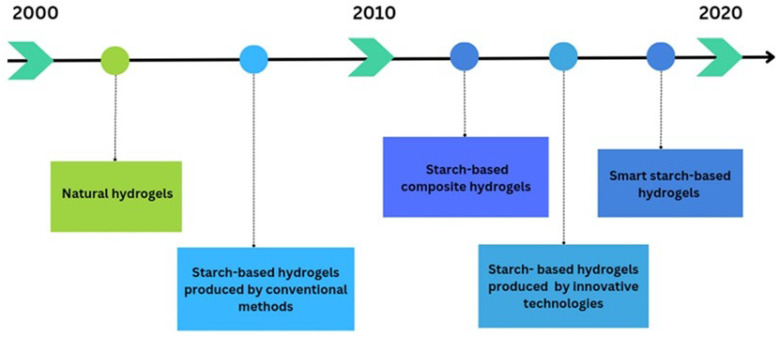
Overview of research on starch-based hydrogels.

**Figure 2 polymers-16-01991-f002:**
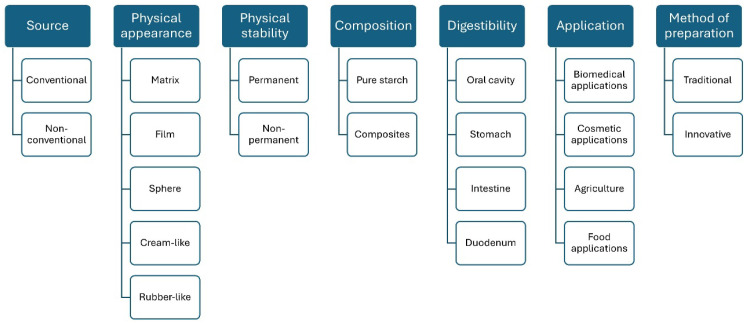
Classification of starch-based hydrogels according to different bases.

**Figure 3 polymers-16-01991-f003:**
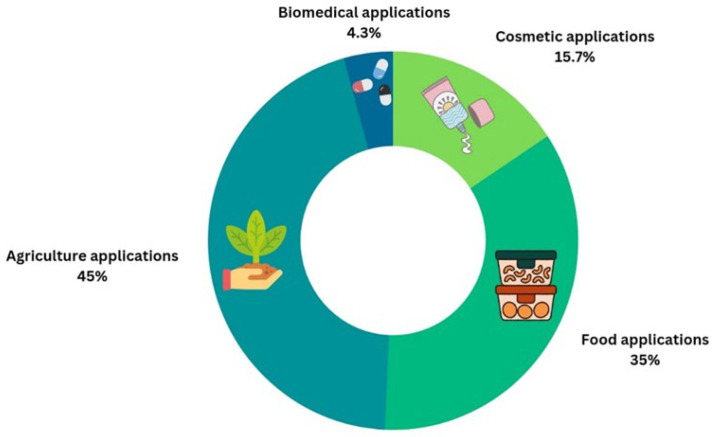
Applications of starch-based hydrogel in the past 10 years (according to Google Patents).

**Figure 4 polymers-16-01991-f004:**
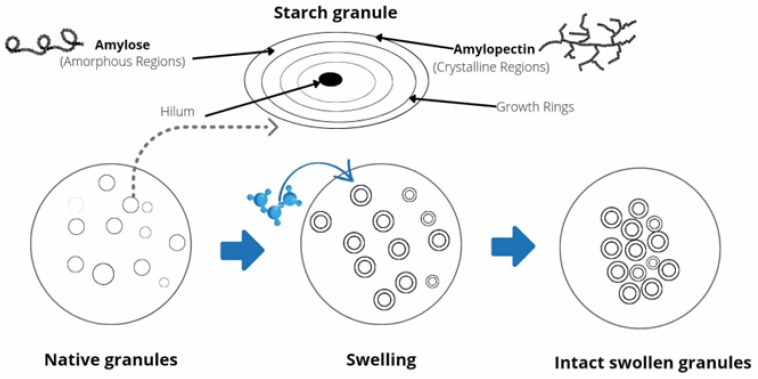
Schematic of starch gelatinization under pressure.

**Figure 5 polymers-16-01991-f005:**
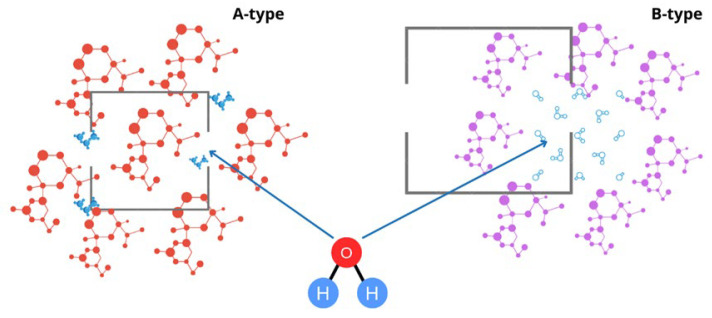
A-type and B-type crystallite conformations.

**Table 1 polymers-16-01991-t001:** Findings from the literature and patents for hydrogel products manufactured from polymeric starch sources.

Patent Number	Year of Publication	Application
US 6,958,157 B1	2005	Disinfectant cleaners
US 7459,501 B2	2008	Agriculture applications
US 2010/0331232 A1	2010	Biomedical applications
EP 2 548 448 A1	2012	Cosmetic applications
US 2012/014 1551A1	2012	Drug delivery device
US 2014/0100111A1	2014	Seed coating hydrogels
US 2015/0250733 A1	2015	Oral drug delivery formulations
US 2020/0138680 A1	2020	Gel-type cosmetic formulations
US2020/0040144 A1	2020	Smart hydrogels from nanometer starch particles
US 10,625,108 B2	2020	Water-retaining agent particles
US 2021/0361570 A1	2021	Innovative applications
CN 114213716 B	2022	Hemostatic dressing
CN 111154039 B	2022	Water-retaining agent particles

**Table 2 polymers-16-01991-t002:** The literature’s findings on the synthesis of starch-based hydrogels for agricultural applications.

Synthesis Method	Gelatinization Conditions	Main Findings	References
Grafting copolymerization	85 °C for 30 min	Hydrogel positively affected seed germination and plant growth.	[112]
Cross-linked starch and cellulose polymer complexes	80 °C for 45 min	Hydrogels showed promise as soil amendments; potato starch hydrogel outperformed the others.	[113]
Biopolymer hydrogel with microwave assistance	70 °C	Stable hydrogels were synthesized and generally degraded in soil.	[114]
Carbendazim-loaded hydrogels	85 °C for 30 min	The hydrogel demonstrated a high-water absorption capacity for soil.	[12]
Cassava starch-graft-poly(acrylamide) copolymers	45 °C for 120 min	Hydrogel improved soil porosity, water retention, nutrient levels, and biological properties, fostering enhanced plant growth.	[115]
Polymerization of AAm and starch	65 °C for 3 h	The synthesis hydrogels showed high water retention, slow atrazine release, and improved soil health post-degradation.	[116]
Ionotropic cross-linking of Cs	76 °C	The synthesis hydrogels were suggested as excellent candidates to be used for the controlled release of fertilizer.	[117]
Graft polymerization with acrylonitrile	78 °C for 10 min	The obtained hydrogel improved water and saline absorbencies compared to native starch-based hydrogel.	[118]
Starch-chitosan hydrogel with atrazine	80 °C for 30 min	The hydrogels loaded with atrazine increased the hydrogel’s thermal stability.	[119]
Superabsorbent hydrogels produced by grafting acrylic acid and itaconic acid.	85 ± 5 °C	Synthetic hydrogels had good biodegradability and great potential for use in agriculture.	[14]
Starch-based smart hydrogel from rice-cooked wastewater	80 °C	Significant water absorption capability, effective for seed germination and growth.	[120]

**Table 3 polymers-16-01991-t003:** Cross-linker agents used in starch-based hydrogel production.

Cross-Linking Agents	Reference
Citric acid	[133]
Glutaraldehyde	[142]
Hydroxypropylate	[151]
Sodium trimetaphosphate	[152]
N,N′-methylene- bisacrylamide	[153]
Aldehyde	[142]
Epichlorohydrin	[135]
Bis-epoxide	[154]
Genipin	[155]
Divinyl Sulfone	[156]

**Table 4 polymers-16-01991-t004:** Sensitivity of various starches under pressure.

Starch Source	X-ray Pattern	Sensitivity under Pressure
Potato, waxy corn, and water yam	B	High
Peanut, amaranth	A	Medium
Arrowroot	C	
Corn, wheat, rice waxy rice, oat, and barley	A	Low
Tapioca, smooth pea lentil, mung bean, faba bean, lotus root, and chestnut	C	

**Table 5 polymers-16-01991-t005:** The literature’s main findings on the use of high pressure as a gelatinization process on starch suspensions.

Source	Main Findings	Reference
Corn, rice, and potato starch	Gelatinization was reached by treatments above 500 MPa for 20 min for starch from corn and rice. Potato starch remained unchanged.	[174]
Corn, waxy corn, waxy rice, potato canna, lotus root, tapioca, taro, chestnut, and pea starch	The pressure range in which the gelatinization starts and is completed depends on the starch type. B-type starch (Potato) was more pressure resistant than A- and C-type starches.	[175]
Normal maize, waxy maize, high amylose maize, tapioca, and rice starches	All the starches evaluated showed a complete gelatinization at 690 MPa for 5 and 60 min.	[10]
Potato starch, wheat starch, and tapioca starch	Pressure-induced gelatinization was highly sensitive to process conditions. Potato starch was fully gelatinized at 700 MPa for 15 min at 50 °C.	[176]
Potato starch	Potato starch only showed a complete gelatinization when treated at 1000 MPa for 1 h at room temperature.	[177]
Corn starch	Corn showed a complete gelatinization at 650 MPa for 20 min at 40 °C.	[170]
Rice and waxy rice starch	At 600 MPa for >10 min, rice starches showed a complete gelatinization. Waxy rice gels showed a higher viscosity.	[11]
Rice starch	Rice starch was completely gelatinized at 600 MPa for 30 min.	[178]
Red azuki bean starch	At 600 MPa, red azuki bean starch was fully gelatinized.	[179]
Pea starch	HPP treatments at 600 MPa for 15 min at 25 °C caused a complete gelatinization of the pea starch suspensions.	[180]
Corn and quinoa	At 600 MPa for 5 min and 25 °C, quinoa and corn starch were gelatinized.	[172]
Sweet potato flour	Sweet potato gels were formed at 600 MPa for 15 min at 25 °C.	[181]
Quinoa flour	HPP induced the gelatinization of quinoa flour at 600 MPa for 1 h at 25 °C.	[182]
Potato starch	Potato starch only showed a complete gelatinization when treated at 600 MPa for 15 min at 50 °C.	[8]
Corn, rice, tapioca, and wheat starch	All the starches were completely gelatinized when treated at 600 MPa for 5 and 15 min.	[7]

**Table 6 polymers-16-01991-t006:** The literature’s findings about high pressure as an alternative method to produce starch-based hydrogels.

Source	Process Conditions	Major Findings	Reference
Potato and corn starch suspensions	300 and 700 MPa, 5 and 25 min at 25 °C	The utilization of HPP allowed selective starch modification beneficial for drug formulation and development.	[190]
Tapioca	600 MPa 10, 20, and 30 min at 30, 50, and 80 °C	Obtained tapioca starch HPP hydrogels showed good mechanical and structural properties.	[191]
Corn, waxy corn, amaranth, and sorghum starch	650 MPa for 9 min at 30 °C. Autoclaving	The differences in the matrix morphology, porosity, and the characteristics of the gels were governed by source origin.	[11]
Pea starch	500/600 MPa for 15 min	High-pressure processing at 600 MPa significantly altered the physical properties of starch granules, leading to gelatinization.	[186]
Potato starch	600 MPa for 15 min	Small granule size (<25 μm) and moderate heating at a low temperature (50° C) enhanced gel formation and structural properties.	[8]
Corn, rice, tapioca, and wheat starch	600 MPa for 5 and 15 min	A longer processing time (15 min) at 600 MPa improved hydrogel mechanical properties but affected color negatively.	[7]
Rice starch	600 MPa for 15 min	The encapsulation of green tea extract in rice starch hydrogels led to structured hydrogels and the controlled release of bioactive compounds.	[50]
Rice starch	500 MPa for 20 min	HHP-induced gelatinization of starch resulted in soft gels that have desirable properties in food products intended for patients with dysphagia.	[192]

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
