# Peer review of "A Comprehensive Review on Starch-Based Hydrogels: From Tradition to Innovation, Opportunities, and Drawbacks"

_polymers, 2024, doi:10.3390/polym16141991_

Round 1

Reviewer 1 Report

Comments and Suggestions for Authors

The authors of the article “A comprehensive review on starch-based hydrogels: from tradition to innovation, opportunities, and drawbacks” provide detailed information on the methods of production of hydrogels based on starch and its derivatives, as well as possible practical applications of the obtained materials. The presented data are undoubtedly of great interest for researchers involved in the synthesis of biomaterials and biocomposites based on polysaccharides, for which we would like to thank the authors.

The article contains an introduction, two chapters and conclusions. Unfortunately, there is a small number of illustrations. In our opinion, such reviews should be illustrated in more details. It is desirable to graphically indicate in the text the structure of starch as a mixture of the polysaccharides amylose and amylopectin.

The data on methods of hydrogel production, related to chemical modification of macromolecules or use of various cross-linking agents, should be systematized and presented in the form of a general table. In such a table it will be sufficient to cite the main methods used in practice with the corresponding references.

The topic of creating composites based on starch and its derivatives is slightly discussed in the paper. Including the creation of bionanocomposites using nanoscale particles. A small expansion of this point would greatly enrich this paper.

The authors repeatedly point out the importance of evaluating the mechanical properties of hydrogels, especially by dynamic rheology methods. The phrase “good rheology properties” is repeated several times, but, unfortunately, there is no description of what it means. Probably it is necessary to give real data with indication of rheological methods used in practice, measured physical quantities and their values. At the same time, it is necessary to compare the properties of hydrogels obtained by different methods or at different parameters.

Unfortunately, the influence of the molecular weight of the starting substances on the hydrogels production and their properties, which is a very important parameter, was not noted in this work.

Lines 507-509. The authors point out that HPP has been proposed as a non-thermal technology. At the same time, the temperature of HPP synthesis of hydrogels is the most important parameter affecting the efficiency of the final result. Comments are needed.

Advanced search on sciencedirect.com based on keywords “starch” and “hydrogel” revealed 372 articles since 2020 year. The authors should check the references again and cite the most recent ones.

There are some typos in the text.

References are cited several times in the text in two formats at once, e.g. lines 334-335: « Mitura et al. (2020) [80] ».

Remaining typos in the text do not have a significant impact on its perception.

Logically structured paper «A comprehensive review on starch-based hydrogels: from tradition to innovation, opportunities, and drawbacks» by Katerina Koshenaj and Giovanna Ferrari describes a current scientific questions and therefore can be recommended for publication in journal «Polymers» after discussion and major revisions.

Comments on the Quality of English Language

The phrases “in vitro” and “in vivo” should be written in italics in the text (e.g., lines 304-305). Several repetitions in one sentence are found in the text. The mentioned typos do not affect the article.  

Author Response

The authors of the article “A comprehensive review on starch-based hydrogels: from tradition to innovation, opportunities, and drawbacks” provide detailed information on the methods of production of hydrogels based on starch and its derivatives, as well as possible practical applications of the obtained materials. The presented data are undoubtedly of great interest for researchers involved in the synthesis of biomaterials and biocomposites based on polysaccharides, for which we would like to thank the authors.

The authors are grateful to the reviewer for investing time in comprehensively reading the review and providing valuable suggestions that have contributed to improving the manuscript. The suggested comments have been integrated into the manuscript wherever possible.

1-The article contains an introduction, two chapters and conclusions. Unfortunately, there is a small number of illustrations. In our opinion, such reviews should be illustrated in more details. It is desirable to graphically indicate in the text the structure of starch as a mixture of the polysaccharides amylose and amylopectin.

We thank the reviewer for this very useful suggestion. The structure of amylose and amylopectin is added in Figure 4 and Figure 5 illustrates the structural differences in amylopectin as revealed by X-ray diffraction analysis.

2-The data on methods of hydrogel production, related to chemical modification of macromolecules or use of various cross-linking agents, should be systematized and presented in the form of a general table. In such a table it will be sufficient to cite the main methods used in practice with the corresponding references.

The authors would like to thank the reviewer for this constructive suggestion. A table with the most used cross-linking agents is added in the manuscript.

3-The topic of creating composites based on starch and its derivatives is slightly discussed in the paper. Including the creation of bionanocomposites using nanoscale particles. A small expansion of this point would greatly enrich this paper.

As suggested by the reviewer, the information related to bio nanocomposites is added.

4-The authors repeatedly point out the importance of evaluating the mechanical properties of hydrogels, especially by dynamic rheology methods. The phrase “good rheology properties” is repeated several times, but, unfortunately, there is no description of what it means. Probably it is necessary to give real data with indication of rheological methods used in practice, measured physical quantities and their values. At the same time, it is necessary to compare the properties of hydrogels obtained by different methods or at different parameters.

Hydrogels are distinctive materials known for their remarkable rheological properties, making them suitable for a wide range of applications. These properties are mainly characterized by their viscoelastic behavior, allowing them to exhibit both elastic and viscous responses to deformation. Good rheological properties enable hydrogels to function effectively in their intended roles by balancing elasticity, viscosity, and mechanical stability. Since the mechanical properties of hydrogels are inherently tied to their specific applications, there are no universal standards. Instead, hydrogels are tailored to meet peculiar functional requirements, ensuring they perform optimally in various biomedical, industrial, and environmental contexts.

5-Unfortunately, the influence of the molecular weight of the starting substances on the hydrogels production and their properties, which is a very important parameter, was not noted in this work.

 We thank the reviewer for this comment. However, the overall molecular weight of starch as the main component used for starch-based hydrogel production represents the sum of the molecular weights of its two main components, namely amylose, and amylopectin. The ratio of amylose/amylopectin in starch determines its overall molecular weight and functional properties and, as discussed in the manuscript, is one of the main factors influencing the occurrence of starch gelatinization.

6-Lines 507-509. The authors point out that HPP has been proposed as a non-thermal technology. At the same time, the temperature of HPP synthesis of hydrogels is the most important parameter affecting the efficiency of the final result. Comments are needed.

We thank the reviewer for this comment. HPP (High-Pressure Processing) is generally classified as a non-thermal technology because the temperature increases during pressure treatments are usually negligible. The primary effects of HPP on microorganisms, enzyme inactivation, and other processes are due to the increase in pressure rather than temperature. This distinguishes HPP from traditional thermal treatments, where temperature plays a central role. However, temperature can become a critical parameter in synthesizing hydrogels using HPP. In these cases, controlling the temperature is crucial for achieving the desired hydrogel properties. If thermal properties are utilized during HPP for hydrogel synthesis, the main advantage of HPP as a non-thermal technology might be compromised, leading to inefficiencies or suboptimal results.

7-Advanced search on sciencedirect.com based on keywords “starch” and “hydrogel” revealed 372 articles since 2020 year. The authors should check the references again and cite the most recent ones.

We would like to thank the reviewer for this valuable comment. A double check is done for the increased number of articles for starch-based hydrogels over time. Moreover, the information is kept up-to-date to ensure that every last paper is quoted in the manuscript.

There are some typos in the text.

References are cited several times in the text in two formats at once, e.g. lines 334-335: « Mitura et al. (2020) [80] ».

Remaining typos in the text do not have a significant impact on its perception.

Logically structured paper «A comprehensive review on starch-based hydrogels: from tradition to innovation, opportunities, and drawbacks» by Katerina Koshenaj and Giovanna Ferrari describes a current scientific questions and therefore can be recommended for publication in journal «Polymers» after discussion and major revisions.

The phrases “in vitro” and “in vivo” should be written in italics in the text (e.g., lines 304-305). Several repetitions in one sentence are found in the text. The mentioned typos do not affect the article.  

We thank the reviewer for the comments. The manuscript has been reviewed and improved to fix any typos and make adjustments to the reference format.

Reviewer 2 Report

Comments and Suggestions for Authors

Dear Authors,

I have completed the review of your manuscript titled "A Comprehensive Review on Starch-based Hydrogels: From Tradition to Innovation, Opportunities, and Drawbacks". However, I would like to recommend some revisions before further consideration.

1)               The manuscript offers a comprehensive overview of the effects of high-pressure processing (HPP) on starch gelatinization and the production of starch-based hydrogels. However, it would be beneficial to begin the manuscript with a clearly stated research objective or hypothesis. What specific research questions or problems does this review aim to address?

2)               The manuscript mentions the classification of starches based on their response to pressure-induced swelling, but it would be helpful to include a summary or classification table that categorizes starches according to their behavior under HPP. This would assist readers in understanding the characteristics and responses of different types of starch.

3)               The review explores the influence of amylose content on starch swelling and gel formation under high pressure. Please provide further details on the mechanisms through which amylose affects these processes. Are there specific analytical techniques or measurements that can be utilized to quantify amylose-lipid interactions and their impact on starch gelatinization?

4)               The manuscript briefly touches upon the impact of high pressure on starch digestibility, highlighting conflicting findings in recent research. Please provide more in-depth information on these contradictory results and discuss potential factors that may contribute to the divergent outcomes. It would be valuable to explore the implications of these findings for the application of starch-based hydrogels in biomedical or food contexts.

5)               While the manuscript acknowledges the potential of non-conventional sources of starch for hydrogel production, it does not delve into specific examples or case studies. Please provide some instances of non-conventional starch sources that have been investigated for hydrogel formation. Additionally, what challenges are associated with the extraction and purification processes for these non-conventional starches, and how can these challenges be addressed?

6)               The manuscript suggests the need for further investigations to fully comprehend the gelation mechanism under high pressure and the effects of processing variables and starch properties on the production of starch-based hydrogels. Please propose specific research directions or experimental approaches that could address these knowledge gaps. Are there any emerging techniques or technologies that could enhance our understanding of starch gelation under pressure?

7)               The manuscript briefly mentions the reduced rheological properties of starch-based HPP hydrogels compared to those obtained through chemical methods. Please provide a more detailed explanation of the rheological properties that are particularly important for the industrial application of starch-based hydrogels. Furthermore, what strategies or additives could be employed to enhance the structural and mechanical properties of starch-based hydrogels? Please provide examples or references, if possible.

8)               The manuscript mentions the necessity of standardizing the physical and microbiological stability of starch-based HPP hydrogels to optimize processing conditions and facilitate their use in various industrial sectors. Please discuss any existing standards or guidelines that can serve as references for assessing the stability of such hydrogels. Are there any specific challenges associated with the stability of starch-based hydrogels, and how can these challenges be addressed?

9)               The manuscript mentions the goal of scaling up the preparation process of starch-based HPP hydrogels for widespread industrial use. Please provide insights into the scalability of HPP for hydrogel production. What are the key considerations and challenges when transitioning from laboratory-scale to industrial-scale production? Are there any examples or case studies of successful scaling-up efforts that can be referenced?

10)            Lastly, the manuscript briefly mentions the application of starch-based hydrogels in various industrial sectors. Please provide specific examples or applications where starch-based hydrogels have demonstrated promise. Are there any notable challenges or limitations associated with their use in these sectors that should be addressed?

Author Response

The authors are grateful to the reviewer for investing time in comprehensively reading the review and providing valuable suggestions that have contributed to the improvement of the manuscript. Where possible, the suggested comments have been incorporated into the manuscript.

1)               The manuscript offers a comprehensive overview of the effects of high-pressure processing (HPP) on starch gelatinization and the production of starch-based hydrogels. However, it would be beneficial to begin the manuscript with a clearly stated research objective or hypothesis. What specific research questions or problems does this review aim to address?

We thank the reviewer for the comment. The review begins with a brief summary that summarizes the overall topic. The research gap is highlighted from lines 22 to 26, along with the review's objective, which is reiterated at the end of the introduction (Line 82-90).

2)               The manuscript mentions the classification of starches based on their response to pressure-induced swelling, but it would be helpful to include a summary or classification table that categorizes starches according to their behaviour under HPP. This would assist readers in understanding the characteristics and responses of different types of starch.

The authors would like to thank the reviewer for this valuable suggestion. A table is added to illustrate the sensitivity of starches under pressure.

3)               The review explores the influence of amylose content on starch swelling and gel formation under high pressure. Please provide further details on the mechanisms through which amylose affects these processes.

The authors would like to thank the reviewer for the valuable suggestions. More details about the influence of amylose content on starch swelling are provided.

Are there specific analytical techniques or measurements that can be utilized to quantify amylose-lipid interactions and their impact on starch gelatinization?

Yes, several analytical techniques and measurements can be utilized to quantify amylose-lipid interactions and their impact on starch gelatinization such as differential Scanning Calorimetry (DSC), X-ray diffraction (XRD), Fourier Transform Infrared Spectroscopy (FTIR), Nuclear Magnetic Resonance (NMR) Spectroscopy.

4)               The manuscript briefly touches upon the impact of high pressure on starch digestibility, highlighting conflicting findings in recent research. Please provide more in-depth information on these contradictory results and discuss potential factors that may contribute to the divergent outcomes. It would be valuable to explore the implications of these findings for the application of starch-based hydrogels in biomedical or food contexts.

As suggested by the reviewer, the authors have provided more information about starch digestibility.

5)               While the manuscript acknowledges the potential of non-conventional sources of starch for hydrogel production, it does not delve into specific examples or case studies. Please provide some instances of non-conventional starch sources that have been investigated for hydrogel formation.

As the reviewer suggested, more information about hydrogel formation using non-conventional starches is provided.

Additionally, what challenges are associated with the extraction and purification processes for these non-conventional starches, and how can these challenges be addressed?

Increasing starch yield during extraction using innovative technologies and, environmentally friendly solvents are important considerations in optimizing the extraction and purification processes for non-conventional starches. Addressing these challenges involves a multidisciplinary approach that combines scientific research, engineering expertise, and economic analysis.

6)               The manuscript suggests the need for further investigations to fully comprehend the gelation mechanism under high pressure and the effects of processing variables and starch properties on the production of starch-based hydrogels. Please propose specific research directions or experimental approaches that could address these knowledge gaps. Are there any emerging techniques or technologies that could enhance our understanding of starch gelation under pressure?

The authors would like to thank the reviewer for the valuable suggestion.

To address the knowledge gaps in understanding starch gelation under high pressure, it is essential to combine advanced analytical techniques, systematic experimental approaches, and interdisciplinary research. The gelation mechanism can be further understood by leveraging emerging technologies and focusing on application-oriented research, which can lead to more efficient production of starch-based hydrogels for various food and non-food applications. Some advanced techniques that can be used are:

Solid-State Nuclear Magnetic Resonance (NMR): To provide detailed information on the molecular dynamics and interactions within the starch matrix.

Atomic Force Microscopy (AFM): To measure the surface properties and mechanical strength of starch hydrogels at the nanoscale.High-Pressure Nuclear Magnetic Resonance (HP-NMR): to study molecular structures and dynamics under high-pressure conditions.

7)               The manuscript briefly mentions the reduced rheological properties of starch-based HPP hydrogels compared to those obtained through chemical methods. Please provide a more detailed explanation of the rheological properties that are particularly important for the industrial application of starch-based hydrogels. Furthermore, what strategies or additives could be employed to enhance the structural and mechanical properties of starch-based hydrogels? Please provide examples or references, if possible.

We thank the reviewer for the comment. The rheological properties of starch-based hydrogels, such as viscosity, shear thinning, elasticity, viscoelasticity, gelation kinetics, yield stress, swelling behavior, and temperature sensitivity, are tailored to meet the demands of specific industrial applications. Each application, whether in food, pharmaceuticals, or biomaterials, requires a particular combination of these properties to ensure optimal performance, processing, and functionality.  

A more detailed explanation of rheological properties is added in the review.

8)               The manuscript mentions the necessity of standardizing the physical and microbiological stability of starch-based HPP hydrogels to optimize processing conditions and facilitate their use in various industrial sectors. Please discuss any existing standards or guidelines that can serve as references for assessing the stability of such hydrogels. Are there any specific challenges associated with the stability of starch-based hydrogels, and how can these challenges be addressed?

We thank the reviewer for the comment. Although there are no universal standards for the assessment of the stability of starch-based HPP hydrogels, existing guidelines from the food, pharmaceutical, agriculture, and cosmetic products can be adapted to evaluate the parameters of interest for these materials. Addressing challenges related to physical stability (gel texture, mechanical strength, and retrogradation), as well as microbiological stability (contamination and shelf life), requires optimizing processing conditions, using advanced analytical techniques, and ensuring regulatory compliance. These steps will facilitate the development and application of starch-based hydrogels across various industrial sectors.

9)               The manuscript mentions the goal of scaling up the preparation process of starch-based HPP hydrogels for widespread industrial use. Please provide insights into the scalability of HPP for hydrogel production. What are the key considerations and challenges when transitioning from laboratory-scale to industrial-scale production? Are there any examples or case studies of successful scaling-up efforts that can be referenced?

The authors would like to thank the reviewer for this valuable comment. To increase the production of starch-based HPP hydrogels for industrial use, one must address challenges such as equipment capacity, maintaining process consistency, material handling, economic feasibility, and regulatory compliance. Pilot studies and process optimization are critical steps toward achieving scalable and economically viable HPP hydrogel production. Moreover, phase separation is a critical challenge in scaling up the production of starch-based HPP hydrogels. Addressing this issue involves careful consideration of material compatibility, processing conditions, and formulation optimization.

10)            Lastly, the manuscript briefly mentions the application of starch-based hydrogels in various industrial sectors. Please provide specific examples or applications where starch-based hydrogels have demonstrated promise. Are there any notable challenges or limitations associated with their use in these sectors that should be addressed?

 The authors would like to thank the reviewer for this comment.  However, Lines 262-420 are devoted to listing four primary applications of starch-based hydrogels, along with their benefits, drawbacks, challenges, and future outlooks.

Reviewer 3 Report

Comments and Suggestions for Authors

The manuscript is well written and comprehensive. Some minor issues should be addressed:

1. The year of publication should not be placed when citing in the text according to editorial requirements, it is suggested to remove.

2. Please check double spaces, it seems they are in lines 315, 399

3. Line 626 - please remove space after []

Author Response

The authors are grateful to the reviewer for investing time in comprehensively reading the review and providing valuable suggestions that have contributed to the improvement of the manuscript. The adjustments are made in the manuscript based on the reviewer's suggestions.

The manuscript is well written and comprehensive. Some minor issues should be addressed:

  1. The year of publication should not be placed when citing in the text according to editorial requirements, it is suggested to remove.
  2. Please check double spaces, it seems they are in lines 315, 399
  3. Line 626 - please remove space after []

Round 2

Reviewer 1 Report

Comments and Suggestions for Authors

All comments identified in the previous round have been addressed. Comments on some questions have been given. This article may be proposed for publication. Nevertheless, we would like to ask the authors once again to specify the references to actual works on this topic. 

One of the key words is "limitations". We would like to ask the authors to elaborate on the limitations of the methods of obtaining hydrogels based on starch and its derivatives. 

Author Response

All comments identified in the previous round have been addressed. Comments on some questions have been given. This article may be proposed for publication. Nevertheless, we would like to ask the authors once again to specify the references to actual works on this topic.

We thank the reviewer again for the constructive suggestions that allowed us to improve the quality of our work. The references of the actual work on the topic are specified in the manuscript.

One of the keywords is "limitations". We would like to ask the authors to elaborate on the limitations of the methods of obtaining hydrogels based on starch and its derivatives.

 We thank the author for the comment. The keyword "limitations" highlights that starch-based hydrogels have drawbacks due to various factors, such as preparation methods and reduced mechanical properties. Specifically, the limitations of the methods used to produce starch-based hydrogels are discussed in lines 519-522. One of the most promising strategies to overcome these limitations is the use of high-pressure processing (HPP) for effective starch modification and gel formation. These limitations are reiterated in the conclusions, where future perspectives are also outlined.

Reviewer 2 Report

Comments and Suggestions for Authors

Dear Authors,

I appreciate your consideration of my comments and your revision, which makes your work clearer and more relevant. I have recommended the publication of your article as is.

Author Response

Dear Reviewer,

Thank you very much for your thoughtful comments and positive feedback. We are grateful for your recommendation to publish our article as is. Your insights have been invaluable in enhancing the clarity and relevance of our work.